# Synthesis of Chitosan-Polyvinyl Alcohol Biopolymers to Eliminate Fluorides from Water

**DOI:** 10.3390/biom10010156

**Published:** 2020-01-18

**Authors:** Cecilia Judith Valdez-Alegría, Rosa María Fuentes-Rivas, José Luis García-Rivas, Rosa Elvira Zavala Arce, María de la Luz Jiménez Núñez, Beatriz García-Gaitán

**Affiliations:** 1Tecnológico Nacional de México/Instituto Tecnológico de Toluca, Av. Tecnológico s/n, Colonia Agrícola Bellavista, Metepec 52149, Estado de México, Mexico; cvaldeza@hotmail.com (C.J.V.-A.); joseluisgarciarivas279@gmail.com (J.L.G.-R.); rzavalaa@toluca.tecnm.mx (R.E.Z.A.); luzjimenez3103@yahoo.com.mx (M.d.l.L.J.N.); 2Facultad de Geografía, Universidad Autónoma del Estado de México, Cerro de Coatepec s/n, Ciudad Universitaria, Toluca 50110, Estado de México, Mexico

**Keywords:** fluoride adsorption, adsorption isotherm, adsorption kinetics, thermodynamic parameters

## Abstract

The fluoride content in groundwater varies depending on geological configuration. Fluoride problems tend to occur in places where these minerals are most abundant in rocks. The objective of the present work was to synthesize four biopolymers based on chitosan-polyvinyl alcohol (Ch-PVA) cross-linked with sodium tripolyphosphate pentabasic (TPP) and ethylene glycol diglycidyl ether (EGDE) and determine their ability to remove fluoride from water. The characterization of the Ch-PVA beads was performed by way of Scanning Electron Microscopy (SEM) and Fourier-transform infrared spectroscopy (FTIR). The percentage of humidity and the point of zero charge were determined. The Ch-PVA beads showed a surface area of 63.87 m^2^ g^−1^, a pore size of 7.6 nm, a point of zero charge of 7.4, and 98.6% humidity. The kinetic adsorption study was adjusted to the pseudo-second-order model and the adsorption equilibrium data were adjusted to the Freundlich adsorption isotherm, showing a maximum fluoride adsorption capacity of 12.64 mg g^−1^ at pH 7 and 30 °C, for the beads of Ch-PVA-NaOH-TPP. According to the thermodynamic parameters: −∆*G*°, +∆*H*° and −∆*S*°, fluoride adsorption is spontaneous, endothermic in nature and there is no random energy change in the solid/liquid interface during the adsorption process.

## 1. Introduction

During the wear and circulation of water in rocks and soils, fluoride is leached and dissolved in groundwater and thermal gases. The fluoride content of groundwater varies greatly depending on the geological configuration and the types of rocks [1,2]. The groundwater in areas containing fluoride-enriched minerals may contain up to 10 to 2800 mg L^−1^ of fluoride [3]. Fluorite, apatite, cryolite, and micas are the most important fluorine-containing minerals. Therefore, fluoride problems tend to occur in places where these minerals are most abundant in host rocks. The consumption of water with fluoride content above 1.5 mg L^−1^ (upper limit recommended by the World Health Organization) can cause damage to the brain and thyroid, endocrine disruption, reproductive decline, immunopathy, decreasing intelligence quotient (IQ), effects on hypertension, as well as dental, skeletal and non-skeletal fluorosis [3]. Fluoride is one of the water-soluble ions that originate from natural and artificial sources, e.g., the industrial effluents discharged from plants producing aluminum and steel, semiconductors, and glass [3,4,5,6,7]. Countries like Argentina, China, India, Kenya, Mexico, South Africa, Tanzania, and Uganda [1,2,8], suffer from fluorosis because of water consumption from wells. Due to the high concentrations of fluoride reported in water, it has been necessary to develop technologies that help eliminate fluoride from groundwater [2].

Among the methods that have been used for the removal of fluoride from water are ion exchange resins, coagulation, chemical precipitation, electrochemical treatment, among others. However, the high cost and poor efficiency of these methods designed to reduce the concentration of fluoride in water have made them infeasible [3,8,9,10,11,12,13]. The fluoride removal process based on the adsorption of biomaterials: alumina, minerals, and metal oxides, clays, metal ion nanoparticles [6,14,15,16,17], polypyrrole, chitosan and chitin [18,19,20,21,22], has been widely accepted due to its capacity for the removal of fluorides in water. The adsorption has peculiar advantages such as low cost, simplicity of design and operation, large sorbent availability, ability to treat a pollutant at a high concentration, insensitivity to toxic substances, and high effectiveness [6]. Chitosan is one of the most promising biopolymers. It is biodegradable, abundant in nature, a good chelating agent, biocompatible, mechanically strong and it has good filming properties. It has been studied in the treatment of drinking water and wastewater, particularly for the removal of heavy metals; leading to its being considered the best option for its low cost and efficiency in the elimination of fluoride [23,24,25] Chitosan is sensitive to pH and to resolve this problem different substances, such as bentonite, montmorillonite, and granular activated carbon have been successfully used in forming composites with chitosan [6].

At present, different methods have been used to improve the chemical and physical properties of chitosan, mainly increasing its pore size, chemical, mechanical, and thermal resistance [26]. One of the polymers used to improve these properties is polyvinyl alcohol (PVA), in conjunction with chemical cross-linking or cross-linking to improve its resistance in acidic and basic solutions [25,26,27,28]. The cross-linking of chitosan and PVA is a reaction with an agent that leads to the preservation of the biopolymer through the formation of bonds between the chitosan and PVA chains. This process can be carried out through covalent or ionic bonds. Covalent cross-linking is an irreversible chemical reaction in which stable bonds are formed between the cross-linking agent and the polysaccharide chains, allowing the chitosan to remain stable even in a very acidic medium (pH 3) [29].

The cross-linking materials used are EGDE and sodium tripolyphosphate pentabasic (TTP). The EGDE has epoxy groups and during the cross-linking process, the epoxy ring of the compound is broken and, simultaneously, the condensation reaction is carried out with the hydroxyl groups of chitosan and PVA, forming covalent bonds. On the other hand, ionic cross-linking forms electrovalent (ionic) bonds between the cross-linking agent and the chitosan chains. In an aqueous solution, the charge of the cross-linking agent is opposite to the charge of the modified polymer. The electrostatic attraction of the polymer chains to those of the cross-linking agent induces the effect of ionic cross-linking. The most common ionic cross-linking agents include citrates and polyphosphates, such as TPP [29]. TPP is a good cross-linker due to its multivalent properties and its non-toxicity. It contains phosphate groups capable of interacting with chitosan through electrostatic interactions. However, it has been shown that chitosan molecules selectively bind with TPP generating several intramolecular and intermolecular complexes [30]. The objective of this work was to synthesize four materials based on chitosan and PVA, cross-linked with EGDE and TPP, and to test their adsorption capacity of fluoride in different batches of aqueous solution, to observe the effect of pH, moisture percentage, the point of zero charge; as well as the characterization of the material with the best adsorption capacity.

## 2. Materials and Methods

### 2.1. Synthesis of Hydrogel Beads

In the synthesis of four hydrogels formic acid (80–91% pure), ethylene glycol diglycidyl ether (EGDE) (50% pure), sodium hydroxide ACS in beads, low viscosity polyvinyl alcohol (PVA), chitosan (Ch) High density, sodium tripolyphosphate pentabasic (TPP) at 98.0% purity, and deionized water were used. The four chitosan-based hydrogels: Ch-PVA-5EGDE, Ch-PVA-7EGDE, Ch-PVA-NaOH-TPP, and Ch-PVA-TPP were prepared by dissolving Ch at room temperature in a 0.4 M formic acid solution. Subsequently, Low Viscosity PVA was added at a temperature of 50–53 °C, until a homogeneous suspension formed [31]. The previous suspension was dripped with an insulin syringe, in a solution of 1.0 M NaOH with magnetic stirring [32]. Once the drip was finished, the Ch-PVA hydrogel beads were stirred in the same NaOH solution for 2 h for maturation.

The Ch-PVA-EGDE cross-linking reaction: Ch-PVA-5EGDE and Ch-PVA-7EGDE was carried out by adding 25 mL of water for every 7 mL of beads, and 5% and 7% by weight of EGDE respectively. Subsequently, the pH of the beads was adjusted to 12 with NaOH (0.1 M) at 70 °C, in an inert atmosphere of N_2_ for 6 h and with constant stirring [32]. While, for the cross-linking of the Ch-PVA-NaOH-TPP hydrogel, the beads were placed in a volume of 100 mL with 2% TPP by weight, and were constantly stirred at room temperature for 12 h [29,33]. In the hydrogel cross-linking process, Ch-PVA-TPP, the Ch-PVA suspension was dripped directly into 100 mL of 2% TPP solution by weight, and constantly stirred at room temperature. Once the drip was completed, the beads obtained were stirred to allow maturation and cross-linking for 12 h in the same TPP solution [29,33]. After the cross-linking time of the four materials, the beads were washed with deionized water until a pH of 6.5–7 was reached, and then kept refrigerated until use.

### 2.2. Effect of pH on Fluoride Adsorption

The effect of pH on fluoride adsorption was studied by way of the preparation of 10 mg L^−1^ F^−^ concentration solutions. The tests were performed in triplicate, with similar amounts of material. The beads were placed in high-density polyethylene bottles, with 5.0 mL of fluoride solution in an orbital thermo-agitator (Heidolph UNIMAX 1010-Inkubator 1000; Schwabach, Germany) at 30 °C and 200 rpm for 72 h. The pH values of the solutions were adjusted in the range of 3 to 9 by the addition of NaOH (0.1 M) or HCl (0.1 M) solutions. After the contact time had elapsed, the supernatant was separated from the solution, using a plastic sieve and 5.0 mL of TISABII solution which was added to this solution to determine the concentration of F^−^ with a Thermo Scientific Orion-4Star model A329 potentiometer, equipped with a selective ion electrode for fluorides, brand: Cole-Palmer, model: CPI-27504 (Chelmsford, MA, USA), previously calibrated.

### 2.3. Hydrogel Characterization

#### 2.3.1. Moisture Percentage

The moisture content was determined by drying the beads to constant weight. Equal amounts of beads of each type of hydrogel were placed in a vacuum desiccator for 48 h until a constant weight was obtained. The moisture percentage was determined by the weight difference [34]. This analysis was performed in triplicate.

#### 2.3.2. Point of Zero Charge (PZC)

To determine the point of zero charge, 10 mL of a NaCl solution (0.1M) with a pH value between 2.0 and 9.0 units, adjusted with hydrochloric acid solution and/or sodium hydroxide, was used. To the NaCl solution, 40 mg of hydrogel (Ch-PVA-NaOH-TPP) was added and kept under stirring at a temperature of 30 °C for 24 h at 200 rpm, upon completion of the stirring time, the supernatant was removed and finally, the final pH was determined [35,36].

#### 2.3.3. Determination of the Surface Area by the BET Method

The specific surface area of the hydrogel was determined using the BET method (Brunauer, Emmett, and Teller) in a BELSORP-max. surfer area and pore volume metre, equipped with a pretreatment degasser BELPRED-vacllsample, of the BelJapan Inc. Brand (Osaka, Japan). The equipment uses a liquid nitrogen bath, at a temperature −196 °C and a pressure of 78 KPa [37].

#### 2.3.4. Fourier Transform Infrared Spectroscopy

The functional groups present in the hydrogel beads before and after the adsorption of the fluoride were identified by Fourier transform infrared analysis (FTIR with ATR); equipment brand: Varian Agilent, model: 640-IR (Santa Clara, CA, USA). The analysis was carried out by means of 16 scans and 4 cm^−1^ resolution and frequency range of 4000–500 cm^−1^. Prior to the analysis, the samples were lyophilized in a device manufactured by the LABCONCO brand, model: 150711772-J (Kansas, MO, USA), for a period of 24 h.

#### 2.3.5. Scanning Electron Microscopy (SEM)

To know the morphology of the hydrogel beads, before and after the adsorption process, scanning electron microscopy analysis was carried out with a JEOL scanning electron microscope, model: JSM 6619LV (Tokyo, Japan), at an acceleration voltage of 20 KV with 50 and WD 12 mm spot-size. The beads were previously lyophilized and subsequently coated with gold plating for analysis.

### 2.4. Batch Adsorption Studies

#### 2.4.1. Adsorption Kinetics

For the adsorption kinetics tests, 0.04 g of hydrogel beads were placed in polyethylene bottles with 5 mL of fluoride solution (Ci = 10 mg L^−1^), in triplicate, under stirring at 200 rpm, for 0, 5, 10, 15, 20, 30, 40, 50, 60, 90, 120, 150 and 180 min, at 10, 30 and 50 °C and pH 5 and 7. Once the adsorption time elapsed, adsorption capacity was determined. Finally, adjustments were made to the pseudo-first-order, pseudo-second-order and Elovich models using the Origin 2016, software (OriginLab corporation, Northampton, MA, USA). The adsorption capacity of fluoride was calculated from Equation (1).
(1)$$q\;=\;\frac{\left(C_{i}{\;-\;C}_{f}\right)V}{m},$$
where: q = Adsorption capacity (mg g^−1^); C_i_ = Initial concentration of adsorbate (mg L^−1^); C_f_ = Final concentration of adsorbate (mg L^−1^); m = mass of the adsorbent (g).

#### 2.4.2. Adsorption Isotherms

To carry out the adsorption isotherms, 0.04g of hydrogel beads were weighed, 5.0 mL of fluoride solution (C_i_ = 10.0 mg L^−1^) was added and placed in high-density polyethylene bottles, under stirring at 200 rpm for 180 min at three temperatures (10, 30 and 50 °C) and two pH values (5 and 7). Subsequently, the measurement was carried out to determine the equilibrium concentration of fluoride in the solution, finally, the adjustment to the Freundlich, Langmuir and Sipps models was made using the Origin 2016 software (Northampton, MA, USA). The test was performed in triplicate.

#### 2.4.3. Thermodynamic Parameters

The study of thermodynamic parameters is based on the calculation of the change in Gibbs free energy (∆*G*°), change in enthalpy (∆*H*°) and change in entropy (∆*S*°), this, with the purpose of understanding the spontaneity, viability, and nature of the adsorption process, through Equation (2) [38]:
∆*G*° = ∆*H*° − *T*∆*S*°,(2)

Using the Vant Hoff equation:
∆*G*° = − (*K*A),(3)
(*K*A) = ∆*S*° *R* − ∆*H*° *RT*,(4)
*K*A = qm/C,(5)
where: C: concentration in the balance of adsorbed solute, qm: concentration in solute equilibrium in solution, *K*A: equilibrium constant (Freundlich equation), *T*: solution temperature (K), *R*: universal gas constant (8.314 J mol^−1^ K^−1^).

## 3. Results and Discussion

### 3.1. Synthesis of Hydrogel Beads

The synthesized hydrogels were hemispherical in shape, of a semi-hard consistency, resistant to touch and white in color [32,39]. After the cross-linking reaction with EGDE, the beads showed no visible changes in their physical characteristics (color and shape). However, because the hydrogel was chemically cross-linked, it acquired a greater resistance, due to the possible drastic reduction of the mobility segments in the polymer by generating a large number of interconnected chains by the formation of new bonds between them [29,40]. The Ch-PVA, which dripped directly onto the TPP cross-linking agent, exhibiting pore size reduction and poor touch resistance; while, the Ch-PVA suspension cross-linked with TPP, after being dripped into NaOH (1.0 M), did not show significant changes in size, shape, or color. The purpose of generating four different materials was to obtain a material with a good adsorption capacity of fluoride in a wide pH range.

### 3.2. Effect of pH on Fluoride Adsorption

The effect of pH variation on the capacity for adsorption of fluoride of each of the synthesized materials: Ch-PVA-5EGDE, Ch-PVA-7EGDE, Ch-PVA-NaOH-TPP, and Ch-PVA-TPP is shown in Figure 1. Once the adsorption process was carried out, the supernatants separated and the 5 mL of TISABII solution was added, an average pH of 5.6 was recorded, due to the buffering function of the TISABII solution. The adsorption capacity of fluoride in the hydrogels, Ch-PVA-5EGDE, and Ch-PVA-7EGDE, was favored at pH 3 (Figure 1), reaching values of 20 and 25 mg g^−1^, respectively, this may be due to the acidic medium. The existing chitosan in the beads, of both materials, maintains the protonated amino groups along the chain and this facilitates the electrostatic interaction between the polymer chains and the negative charge of the fluoride [20]. The adsorption capacity with the Ch-PVA-TPP material was not significant, concerning the other materials (Ch-PVA-5EGDE, Ch-PVA-7EGDE and Ch-PVA-NaOH-TPP), this could possibly be because the cross-linking material occupied the active sites of chitosan beads [19]. Likewise, it was observed that the hydrogel of Ch-PVA-NaOH-TPP showed greater absorption of fluoride at pH 5, followed by pH 7 and pH 6. The material Ch-PVA-NaOH-TPP, was characterized and used in adsorption tests.

### 3.3. Characterization of Hydrogel Beads

#### 3.3.1. Moisture Percentage, Surface Area, Pore Size and Point of Zero Charge (PZC) of the Ch-PVA-NaOH-TPP Beads

The moisture percentage of the four synthesized materials were Ch-PVA-5EDGE (96.8%), Ch-PVA-7EDGE (97.5%), Ch-PVA-TPP (89.9%) and Ch-PVA-NaOH-TPP (98.6%). According to these results, the four hydrogels are high swelling composites [32,41]. Regarding the surface area, the pore size and point of zero charge of the Ch-PVA-NaOH-TP material, selected for a complete analysis, were 63.87 (m^2^ g^−1^), 7.6 (nm) and 7.4, respectively. A convenient pore classification, was originally proposed by Dubinin in 1992 and, subsequently, officially adopted by the International Union of Pure and Applied Chemistry (IUPAC). This classification is based on the properties that the different pores have according to their dimension in the adsorption processes and are manifested in the adsorption isotherms, according to this consideration the analyzed material can be classified as mesoporous [19,37].

#### 3.3.2. Characterization by Fourier Transform Infrared Spectroscopy

The Ch-PVA materials obtained, characterized by FTIR, before and after the fluoride adsorption process. The spectra of Ch, PVA, and TPP, are present in Figure 2a. In the chitosan spectrum, the band at 3354 cm^−1^ characteristics of the stretching of the -OH group and the signal at 3290 cm^−1^, of the stretching of the -NH group; [42,43,44,45,46,47,48,49,50]. The vibrations at 2920 cm^−1^, 2870 cm^−1^ are characteristic of the stretching of the asymmetric and symmetric CH group, respectively [42,43,44,45,46,50]. The vibration of 1659–1554 cm^−1^ is caused by the stretching of the C=O and N-H groups, characteristic of amino groups; while, the 1589 cm^−1^ signal can also be attributed to the flexural vibration of -NH [42,43,44,45,46,50]. Likewise, the peak at 1416 cm^−1^ is characteristic of the torsion of the CH_2_ group, [50,51]. The signal at 1373 cm^−1^ is due to the symmetrical flexion of CH [45,46]. The peak at 1319 cm^−1^ is due to the tension of the CN group [50,51]. The bands at 1059 and 1026 cm^−1^ are characteristic of the stretching of the CO group and the signal at 890 cm^−1^ is characteristic of the stretching of the COC glycoside group [50,51].

In the PVA spectrum, a wideband centered at 3323 cm^−1^ of stretching corresponding to the OH group is observed [52]. The vibration of 2935–2910 cm^−1^ represents a strong band corresponding to the stretching of the CH and CH_2_ groups [52,53]. The high intensity peak at 1734 cm^−1^ corresponds to the tension of the C=O bond of the acetate groups. The band at 1419 cm^−1^ corresponds to the flexion of the CH_2_ group. The signal at 1373–1338 cm^−1^ corresponds to the flexion of the CH_3_ group. The weak band at 1242 cm^−1^ corresponds to the flexion of the C-OH bond of the alcohols [53]. The vibration at 1140 cm^−1^ is due to the stretching of the C−C and C−O groups [53,54], the 1092 cm^−1^ signal corresponds to the stretching of the C−O group and the 845 cm^−1^ stretch vibration corresponds to the C–C [54]. In the TPP spectrum, a band at 1211 cm^−1^ of stretching corresponding to the group P=O, [29,55,56] is observed. The vibration at 1136 cm^−1^ corresponds to the symmetrical and asymmetric stretching of the PO_2_ groups, [29,56] and the peak at 887 cm^−1^ is related to the stretching vibrations of the POP group [29,56].

Figure 2b shows the spectrum of Ch-PVA-NaOH-TPP before the adsorption process. It shows the signal 3354 cm^−1^ of the chitosan OH group, with a slight increase to 3359 cm^−1^ and the band at 3323 cm^−1^ of the OH group of the PVA loses definition [57]. A slight decrease of 2918 cm^−1^ is observed in the 2920 cm^−1^ signal of the CH group of the chitosan. In the band 1589 cm^−1^ of the NH group a slight reduction in intensity is observed at 1576 cm^−1^. In the bands of 1059–1026 cm^−1^ of the CO group, a very slight displacement is observed at 1061–1026 cm^−1^. Due to the interaction with the PVA. A slight increase to 895 cm^−1^ derived from the interaction between the chitosan and the PVA which is observed at the peak of 890 cm^−1^ of the COC group [58].

With respect to the chitosan cross-linked with TPP, it is observed that the bands at 3354 cm^−1^ (OH) and 3290 cm^−1^ (NH) became wider, increasing to 3359 cm^−1^ and 3292 cm^−1^, indicating a possible external interaction in these groups. That is, given the nature of the cross-linker, it could be a hydrogen bridge between the amine and the oxygen of the tripolyphosphate [56,59]. The weak signal at 1140 cm^−1^ increased slightly at 1148 cm^−1^. This might be due to the PO group which provides evidence for the cross-linking of chitosan [29,60].

In the spectrum of Ch-PVA-NaOH-TPP after the adsorption process, a displacement at 3363 cm^−1^ is observed in the stretch band at 3359 cm^−1^ of the OH group, which could be due to the electrostatic interaction of the fluoride with the OH group [18,23,45,46,61,62]. The 2918 cm^−1^ and 2870 cm^−1^ peaks of the CH groups, were displaced at 2920 cm^−1^ and 2872 cm^−1^ due to the possible interaction of the CH group with the fluoride [63]. The band at 1576 cm^−1^ of the NH group shifted to 1593 cm^−1^ indicating a possible electrostatic interaction between the NH group of the Chitosan and fluoride [10,45,46,62].

#### 3.3.3. Characterization by Scanning Electron Microscopy (SEM)

Figure 3 shows the micrographs of the Ch-PVA beads before and after the adsorption process, in these micrographs, there are well-defined surfaces with ridges, valleys, and pores. In addition, the presence of channels facing in the same direction and channels with a homogeneous and porous surface can be observed. The morphological difference between the outer and inner surface of the hydrogel bead was differentiated. The morphological characteristics of the interior of the bead may be due to the cross-linking process with TPP contrary to Borba et al. (2016) and Shen et al. (2016) [64,65].

### 3.4. Adsorption Studies

#### 3.4.1. Adsorption Kinetics.

According to the data reported in Table 1, the Ho-McKay model is the one that best represents the behavior of the experimental data for the three temperatures (10, 30 and 50 °C) at pH 5 and pH 7 [6,22,66]. This model suggests that the active sites of the adsorbent are heterogeneous, and that there is a chemical adsorption process [20,66]. In Figure 4, the graphs adjusted to the kinetic models at pH 5 and pH 7 can be observed.

#### 3.4.2. Adsorption Isotherm

The data obtained were adjusted to the Freundlich, Langmuir, and Langmuir-Freundlich models by means of a non-linear regression using the Origin 8.1 program. The values of these adjustments are shown in Table 2, the coefficients of determination (R^2^) are very similar, but performing the chi-square test shows that the Freundlich model is the one that best fits the three temperatures (10, 30 and 50 °C) at pH 5 and pH 7, this suggests that in adsorption, a physisorption may be involved in a homogeneous material [22]. Figure 5 show the graphs adjusted to the isothermal models at pH 5 and pH 7.

#### 3.4.3. Thermodynamic Parameters

Thermodynamic calculations of an adsorption process are essential in concluding whether the aforementioned process is spontaneous or not. From the slope and the intersection of the linear curve drawn in KA vs. 1/T, it was possible to obtain the enthalpy and entropy values, respectively [67,68,69]. A negative value of ∆G indicates the viability of the adsorption process. The change in ∆*G*° (initial and final system conditions) is the amount of energy taken into account in the corresponding contributions of entropy and enthalpy. This is in order to predict the spontaneity of the adsorption process under consistent conditions of pressure and temperature. The spontaneity of reactions and processes will depend on the sign of ∆H, the sign of ∆S and the value of temperature [68,69]. The Gibbs free energy value (∆*G*°) was −1452.3 KJ mol^−1^, suggesting that the adsorption process is spontaneous [19,22].

When ∆H is positive, the adsorption process is endothermic and adsorption will be favored by an increase in system temperature. With an increase in temperature, the value of T∆S increases until it exceeds the value of ∆H, therefore ∆G is negative and the process is spontaneous. A positive ∆*S*° value ∆*S*° indicates that the degree of disorder in the solid/liquid interface increases in the adsorption system, and if the temperature rises, the change in enthalpy is favored [38,70]. The change in entropy ∆S is described as the dispersion of energy and matter, that is, the larger the dispersion of energy in a system or the degree of disorder in the solid/liquid interface, the greater the opposite entropy when a negative entropy occurs [70]. The changes in enthalpy (∆*H*°) and entropy (∆*S*°) were 1.62 KJ mol^−1^ and −299.7 KJ mol^−1^, the positive value ∆*H*° suggests that the adsorption process is endothermic in nature [19,22] and the negative value of ∆*S*° confirms that there is no increase in randomness in the solid/liquid interface during adsorption [22,38,71].

### 3.5. Fluoride Adsorption Mechanism

The presence of the amino group in the chitosan was confirmed by FTIR analysis. The nitrogen in the amino group of chitosan yields electrons and is responsible for selective chelation with metal ions. The OH group present can bind or release protons depending on the pH of the initial solution, resulting in the development of surface charges. Under acidic conditions, sites with more positive charge yield that result in a higher capacity for sorption of fluoride, which is evident from the results obtained, where a maximum capacity for sorption of fluoride at pH 5 is observed. At a pH greater than 6, a significant decrease in fluoride sorption capacity [23] was observed. Therefore, it can be concluded that the removal of fluoride by chitosan is governed by electrostatic attraction, so the following reaction mechanism is possible which confirms the decrease in pH of the solution after the adsorption process.
(6)$${Q-\mathrm{NH}}^{+}\;+\;F^{-}\;⇌{\;Q-\mathrm{NHF}\;+\;H}^{+}$$

## 4. Conclusions

Knowing the current problem regarding the contamination of groundwater destined for human consumption and the negative effects on health due to the high fluoride content, this study led to the synthesis of four chitosan-based hydrogels, the Ch-PVA-NaOH-TPP, material showed better results in contact tests in aqueous solution, with a higher percentage of porosity and humidity (98.6%). The highest fluoride adsorption (14.3 mg g^−1^) was at pH 5. The experimental data best fit the Ho McKay model at pH 5 and 50 °C, indicating the heterogeneous nature of the adsorption process. Besides, the adjustment to the Freundlich model indicates multilayer adsorption at pH 5 and 30 °C. According to thermodynamic parameters the adsorption process is spontaneous (∆*G*° < 0), endothermic (∆*H*° > 0) and occurs with a decrease in entropy (∆*S*° < 0). The electrostatic interaction between fluoride and protonated chitosan amino groups was responsible for the affinity of the hydrogel and fluoride. Therefore, this feasible, inexpensive, and easily obtainable biomaterial can affect the removal of fluoride in aqueous solution.

## Figures and Tables

**Figure 1 biomolecules-10-00156-f001:**
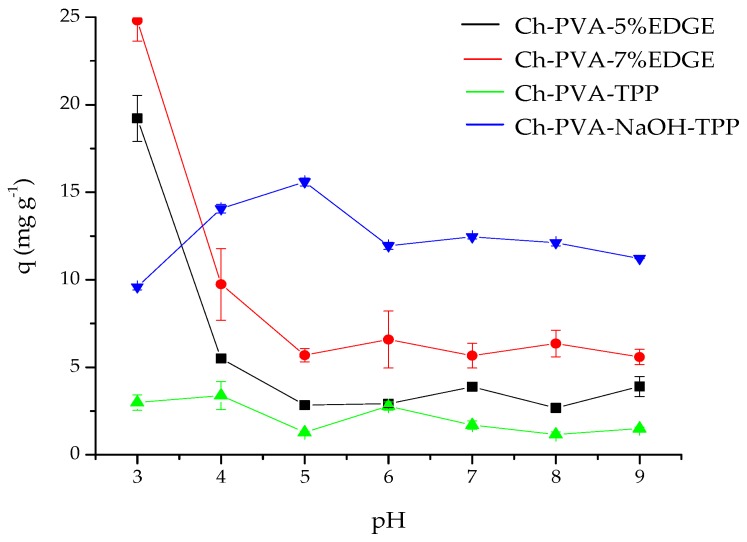
Effect of pH on the adsorption capacity of fluoride of synthesized materials: Q-PVA-5EGDE, Ch-PVA-7EGDE, Ch-PVA-NaOH-TPP, and Ch-PVA-TPP.

**Figure 2 biomolecules-10-00156-f002:**
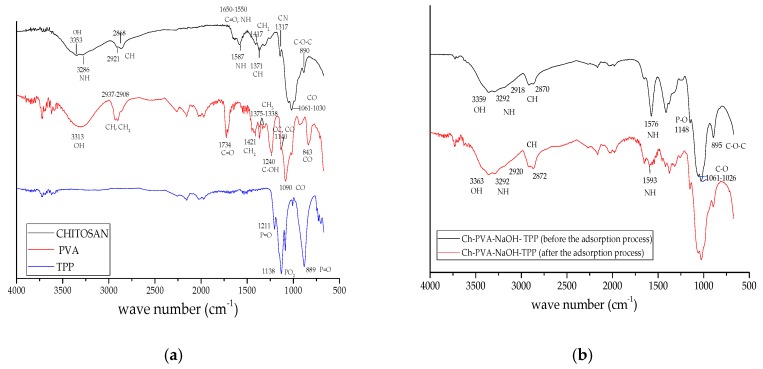
(**a**) FTIR spectra of chitosan-polyvinyl alcohol (Ch-PVA), and tripolyphosphate pentabasic (TPP); (**b**) FTIR spectra of Ch-PVA-NaOH-TPP before and after adsorption.

**Figure 3 biomolecules-10-00156-f003:**
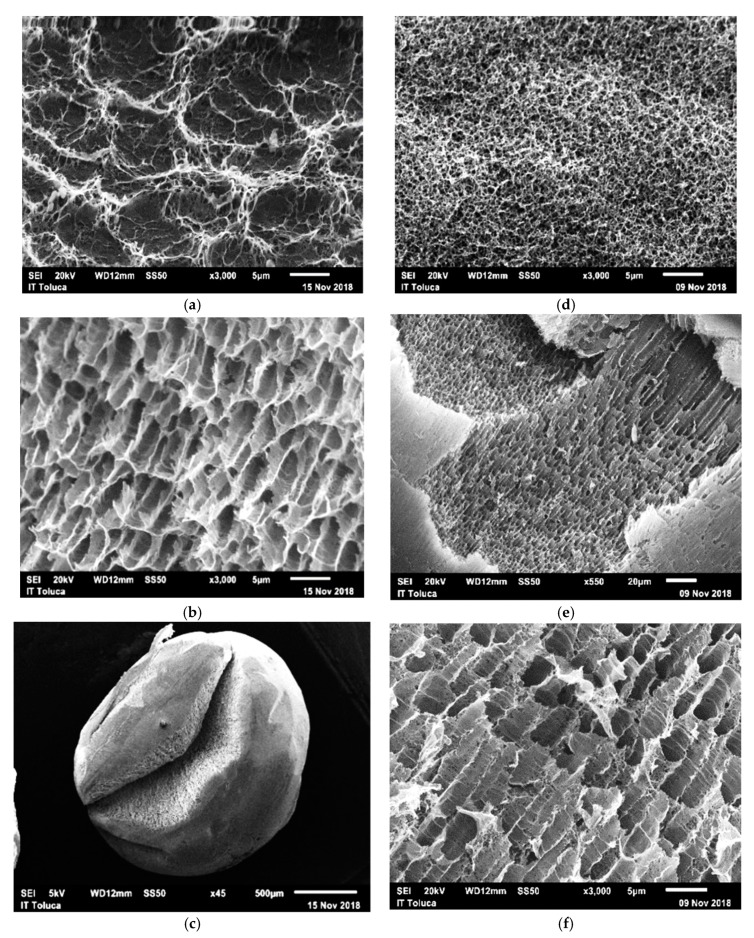
Micrographs of the Ch-PVA-NaOH-TPP hydrogel; before the adsorption process (**a**) surface with ridges, valleys, and pores; (**b**) the presence of channels with porous surface walls facing the same direction; (**c**) morphological difference between the external and internal surface of the hydrogel bead. Hydrogel micrograph Ch-PVA-NaOH-TPP; after the adsorption process (**d**) morphological difference between the external and internal surface of the hydrogel bead; (**e**) channels facing in the same direction and pores in the surface wall of the channels; (**f**) homogeneous and porous surface.

**Figure 4 biomolecules-10-00156-f004:**
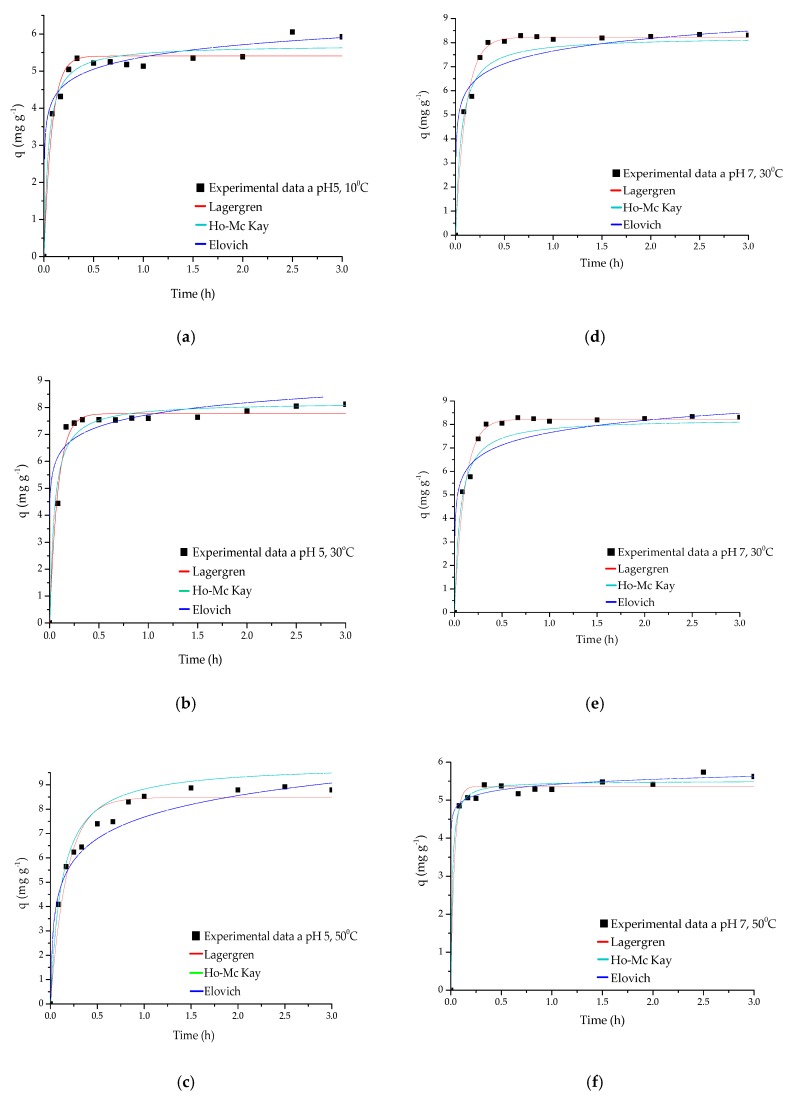
Adsorption kinetics (**a**) Q-PVA-NaOH-TPP (pH 5, T = 10 °C); (**b**) Ch-PVA-NaOH-TPP (pH 5, T = 30 °C); (**c**) Ch-PVA-NaOH-TPP (pH 5, T = 50 °C); (**d**) Ch-PVA-NaOH-TPP (pH 7, T = 10 °C); (**e**) Ch-PVA-NaOH-TPP, (pH 7, T = 30 °C) and (**f**) Ch-PVA-NaOH-TPP, (pH 7, T = 50 °C).

**Figure 5 biomolecules-10-00156-f005:**
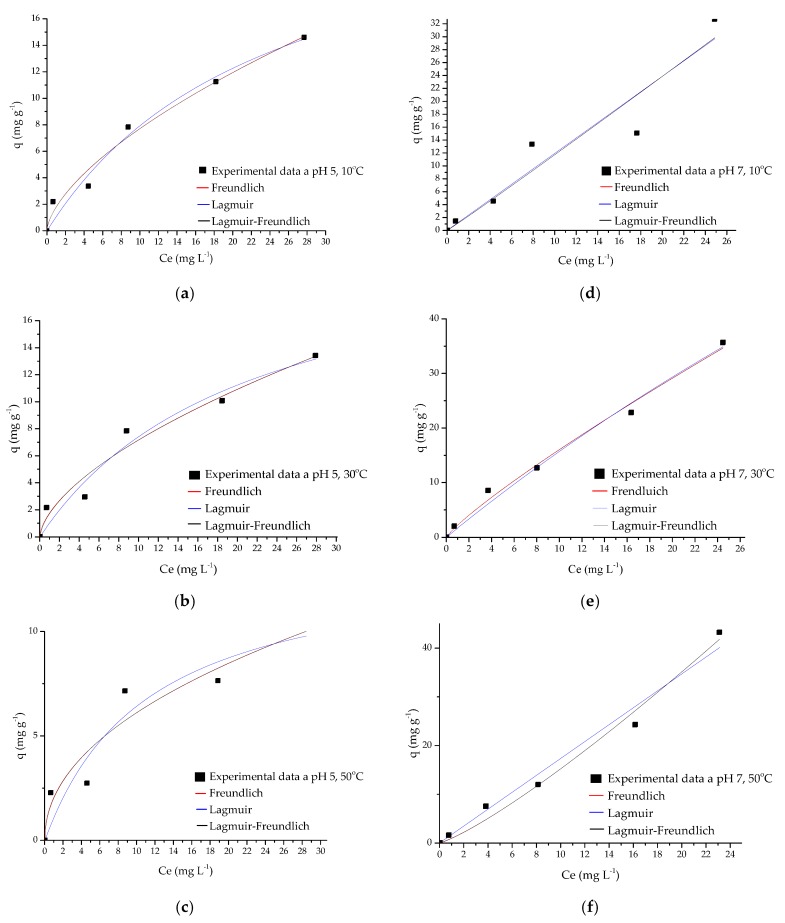
Adsorption isotherm (**a**) Ch-PVA-NaOH-TPP (pH 5, T = 10 °C); (**b**) Ch-PVA-NaOH-TPP (pH 5, T = 30 °C); (**c**) Ch-PVA-NaOH-TPP (pH 5, T = 50 °C); (**d**) Ch-PVA-NaOH-TPP (pH 7, T = 10 °C); (**e**) Ch-PVA-NaOH-TPP (pH 7, T = 30 °C) and (**f**) Ch-PVA-NaOH-TPP (pH 7, T = 50 °C).

**Table 1 biomolecules-10-00156-t001:** Summary of adjustment of experimental data to kinetic models.

**T (°C)**	**q_exp_** **(mg g^−1^)**	**Lagergren**					**Ho**				**Elovich**		
		**pH 5**											
		**K_1_** **(min^−1^)**	**q_e_** **(mg g^−1^)**	**R^2^**	**SSE**	**K_1_** **(min^−1^)**	**q_e_** **(mg g^−1^)**	**R^2^**	**SSE**	**A** **(mg g^−1^ h^−1^)**	**Β** **(g mg^−1^)**	**R^2^**	**SSE**
10	5.35	12.53	5.41	0.96	13.28	4.19	5.69	0.98	0.03	3.7 × 10^4^	2.1027	0.97	0.03
30	7.56	11.81	7.78	0.99	13.28	2.76	8.18	0.97	0.06	9.2 × 10^4^	1.55	0.93	0.11
50	8.30	5.46	8.57	0.96	16.50	0.998	9.13	0.99	0.01	4.0 × 10^4^	1.45	0.87	0.18
**T (°C)**	**q_exp_** **(mg g^−1^)**	**Lagergren**					**Ho**				**Elovich**		
		**pH 7**											
		**K_1_** **(min^−1^)**	**q_e_** **(mg g^−1^)**	**R^2^**	**SSE**	**K_1_** **(min^−1^)**	**q_e_** **(mg g^−1^)**	**R^2^**	**SSE**	**α** **(mg g^−1^ h^−1^)**	**Β** **(g mg^−1^)**	**R^2^**	**SSE**
10	5.52	15.653	5.36	0.99	12.84	7.149	5.23	0.99	0.02	3.3 × 10^8^	3.96	0.96	0.04
30	8.01	9.41	8.22	0.98	13.09	2.04	8.68	0.98	0.3	1.7 × 10^4^	1.2515	0.94	0.08
50	5.40	26.80	5.37	0.98	12.25	14.28	5.49	0.99	0.01	1.3 × 10^11^	5.05	0.99	0.01

**Table 2 biomolecules-10-00156-t002:** Summary of adjustment of experimental data to adsorption isothermal models.

**T (°C)**	**q_exp_** **(mg g^−1^)**	**Freundlich**				**Langmuir**				**Langmuir-Freundlich**				
		**pH 5**												
		**k***	**N**	**R^2^**	**X^2^**	**b (L mg^−1^)**	**q_emax_** **(mg g^−1^)**	**R^2^**	**X^2^**	**Ks** **(mg^1−n^ L^3ng^)**	**n**	**q_m_**	**R^2^**	**X^2^**
10	14.61	1.80	1.58	0.98	0.92	0.04	27.09	0.99	328.5	7.5 × 10^−5^	0.64	771	0.98	16.9 × 10^5^
30	13.40	1.77	1.65	0.97	0.74	0.05	23.16	0.96	132.5	5.3 × 10^−5^	0.62	749	0.97	15.4 × 10^5^
50	10.20	2.06	2.12	0.93	1.15	0.09	13.63	0.92	151.2	3.1 × 10^−6^	0.48	859	0.93	51.2 × 10^5^
**T (°C)**	**q_exp_** **(mg g^−1^)**	**Freundlich**				**Langmuir**				**Langmuir-Freundlich**				
		**pH 7**												
		**k***	**N**	**R^2^**	**X^2^**	**b** **(L mg^−1^)**	**q_emax_ (mg g^−1^)**	**R^2^**	**X^2^**	**Ks** **(mg^1−n^ L^3ng^)**	**n**	**q_m_**	**R^2^**	**X^2^**
10	32.69	1.10	0.98	0.92	4.28	2.14 × 10^−5^	4 × 10^4^	0.92	11.9 × 10^3^	5.30 × 10^−5^	1.06	2.7 × 10^6^	0.92	16.9 × 10^5^
30	35.65	1.68	1.12	0.99	0.74	5.84	0.0046	0.99	407.2	4.61 × 10^−5^	0.89	1.2 × 10^6^	0.99	4.65 × 10^5^
50	45.24	0.84	0.87	0.99	3.24	1.68 × 10^−5^	8.1 × 10^6^	0.98	411.9	1.45 × 10^−4^	1.15	2.2 × 10^6^	0.99	2.64 × 10^6^

* k: [(mg g^−1^) (L mg^−1^)^1/n^].

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
