# Peer review of "Synthesis of Chitosan-Polyvinyl Alcohol Biopolymers to Eliminate Fluorides from Water"

_biomolecules, 2020, doi:10.3390/biom10010156_

Round 1
Reviewer 1 Report
Elimination of harmful fluoride from drinking water should be intensively investigated. The article submitted for review could therefore be useful and valuable. Unfortunately, in current form it is unsuitable for publication.
In "Results and Discussion" section there is no discussion. The authors present their own results without reference to those obtained by the others.
Conclusions are not conclusions. This section recapitulates the data obtained. Instead, the reader would like to know the authors' opinion regarding effectiveness and usefulness of the examined biopolymers in fluoride elimination from water. This opinion was simply not formulated.
The literature is out of date to a large extent. Most references (31) have been published between 2009 and 2014. Only fourteen papers come from 2015-2018. None has been published in 2019. At least the following articles published in 2017-2019 have not been taken into consideration:
Fluoride 2019 Dec 6. www.fluorideresearch.online/epub/files/064.pdf [Epub ahead of print] PURIFICATION OF SIMULATED FLUORIDE-CONTAMINATED WATER USING SILVER NANOPARTICLES SYNTHESIZED BY XANTHIUM STUMARIUM AS A BIO-TEMPLATE
Mudassar Iqbal, Muhammad Suleman, Saleem Ullah, Hamida Bibi, Hassan Wahab, Zafar Iqbal, Khadim Muhammad Dawar, Said Wahab, Muhammad Nauman Ahmad
Fluoride 52(4):531-536 2019
APPLICATION OF ELECTROCHEMICAL PROCESSES FOR THE REMOVAL OF FLUORIDE FROM POLLUTED WATER
Atieh Salem, Nezam Mirzaei, Amir Hossein Mahvi, Hossein Akbari, Zeinab Parmoozeh, Davarkhah Rabbani
Fluoride 52(4):546-552 2019
ADSORPTION OF FLUORIDE FROM AQUEOUS SOLUTIONS BY A CHITOSAN/ZEOLITE COMPOSITE
Amir Hossein Mahvi, Ferdos Kord Mostafapour, Davoud Balarak, Aram Dokht Khatibi
Fluoride 52(4):562-568 2019
ADSORPTION OF FLUORIDE FROM AQUEOUS SOLUTION BY EUCALYPTUS BARK ACTIVATED CARBON: THERMODYNAMIC ANALYSIS
Amir Hossein Mahvi, Ferdos Kord Mostafapour, Davoud Balarak
Fluoride 52(4):569-579 2019
REMOVAL OF FLUORIDE FROM AQUEOUS SOLUTION BY NICKEL OXIDE NANOPARTICLES: EQUILIBRIUM AND KINETIC STUDIES
Chinenye Adaobi Igwegbe, Somayeh Rahdar, Abbas Rahdar, Amir Hossein Mahvi, Shahin Ahmadi, Artur Marek Banach
Fluoride 52(3 Pt 2):299-318 2019
MODELING AND OPTIMIZATION OF FLUORIDE ADSORPTION FROM AQUEOUS SAMPLES BY AMMONIUM ALUMINIUM SULFATE USING RESPONSE SURFACE METHODOLOGY (RSM)
Mojtaba Afsharnia, Mahmood Shams, Mehdi Ghasemi, Hamed Biglar, Samira Salari, Asiyeh Moteallemic Gonaba
Fluoride 52(3 Pt 1):231-247 2019
REMOVAL OF FLUORIDE FROM DRINKING WATER BY FREEZING TECHNOLOGY Sara Sadat Hosseini, Amir Hossein Mahvi
Fluoride 51(4)319–327 2018
ADSORPTION OF FLUORIDE ON CHITOSAN IN AQUEOUS SOLUTIONS: DETERMINATION OF ADSORPTION KINETICS
Hamideh Akbari, Sahand Jorfi, Amir Hossein Mahvi, Masoud Yousefi, Davoud Balarak
Fluoride 51(4)355–365 2018
HIGHLY EFFICIENT ADSORPTION OF FLUORIDE FROM AQUEOUS SOLUTIONS BY METAL ORGANIC FRAMEWORKS: MODELING, ISOTHERMS, AND KINETICS Mohamadreza Massoudinejad, Abbas Shahsavani, Bahram Kamarehie, Ali Jafari, Mansour Ghaderpoori, Mostafa M Amini, Afshin Ghaderpoury
Fluoride 51(3)220–229 2018
DETERMINATION OF FLUORIDE BIOSORPTION FROM AQUEOUS SOLUTIONS USING ZIZIPHUS LEAF AS AN ENVIRONMENTALLY FRIENDLY COST EFFECTIVE BIOSORBENT
Amir Hossein Mahvi, Sina Dobaradaran, Reza Saeedi, Mohammad Javad Mohammadi, Mozhgan Keshtkar, Arefeh Hosseini, Mahsa Moradi, Fatemeh Faraji Ghasemi
Fluoride 51(1)34–43 2018
DEFLUORIDATION OF GROUNDWATER IN CENTRAL MEXICO BY ELECTROCOAGULATION
Luna J, Martinez J, Montero C, Muñiz C, Ortiz J, Gonzalez G, Vazquez V, Equihua F
Fluoride 50(2)223–234 2017
THE EQUILIBRIUM, KINETIC, AND THERMODYNAMIC PARAMETERS OF THE ADSORPTION OF THE FLUORIDE ION ON TO SYNTHETIC NANO SODALITE ZEOLITE
Davoud Balarak, Ferdos Kord Mostafapour, Edris Bazrafshan, Amir Hossein Mahvib
Fluoride 50(2)256–268 2017
SYNTHESIS OF NANOCHITOSAN FOR THE REMOVAL OF FLUORIDE FROM AQUEOUS SOLUTIONS: A STUDY OF ISOTHERMS, KINETICS, AND THERMODYNAMICS
Ali Naghizadeh, Habibeh Shahabi, Elham Derakhshani, Fatemeh Ghasemi, Amir Hossein Mahvi
Articles 46 - 53 are not listed in the "References" section.
Minor points:
There is no need to write: "fluoride ion". Fluoride is an ion (ionized form of fluorine).
Line 38: fluorite, apatite and cryolite are the most important fluorine-containing minerals
Author Response
Reviewer 1
Dear reviewer, in attached file we send the correccones and suggested adjustments.
Elimination of harmful fluoride from drinking water should be intensively investigated. The article submitted for review could therefore be useful and valuable. Unfortunately, in current form it is unsuitable for publication.
The introduction has included text referring to the suggested research to highlight the importance of fluoride removal from wáter.
In "Results and Discussion" section there is no discussion. The authors present their own results without reference to those obtained by the others.
In results and discussion, we referred to other authors, we included current references, mainly the references suggested by the reviewer were considered to enrich this section.
Conclusions are not conclusions. This section recapitulates the data obtained. Instead, the reader would like to know the authors' opinion regarding effectiveness and usefulness of the examined biopolymers in fluoride elimination from water. This opinion was simply not formulated.
The conclusions have been modified without neglecting the important data obtained in this study, in recent years there have been enumerated research with biopolymers which have been recognized, for their easy obtaining and low cost, since they are capable of adsorbing ions and / or releasing drugs, pesticides, among others.
The literature is out of date to a large extent. Most references (31) have been published between 2009 and 2014. Only fourteen papers come from 2015-2018. None has been published in 2019. At least the following articles published in 2017-2019 have not been taken into consideration:
The suggested references have been integrated and we include others, is important to mention that there is not a very large number of articles that refer research related whith our research. We appreciate all the suggestions given to enrich this work.
Articles 46 - 53 are not listed in the "References" section.
The articles were included in the document
Minor points:
There is no need to write: "fluoride ion". Fluoride is an ion (ionized form of fluorine).
Line 38: fluorite, apatite and cryolite are the most important fluorine-containing minerals
The change has been made.

Reviewer 2 Report
This paper is an extremely important for ecology in order to eliminate fluoride ions from groundwater. The objective of the present work was to synthesize four biopolymers based on chitosan-polyvinyl alcohol (Ch-PVA) cross-linked with sodium tripolyphosphate pentabasic (TPP) and ethylene glycol diglycidyl ether (EGDE) and determine their ability to remove fluoride ions from water.
The paper is well organized, well argued with specific analytical analysis and has a proper mathematical processing of the involved adsorption/retention processes
I recommend to publish this paper in the present version.
Author Response
Reviewer 2
Dear reviewer, in attached file we send the correccones and suggested adjustments.
This paper is an extremely important for ecology in order to eliminate fluoride ions from groundwater. The objective of the present work was to synthesize four biopolymers based on chitosan-polyvinyl alcohol (Ch-PVA) cross-linked with sodium tripolyphosphate pentabasic (TPP) and ethylene glycol diglycidyl ether (EGDE) and determine their ability to remove fluoride ions from water.
The paper is well organized, well argued with specific analytical analysis and has a proper mathematical processing of the involved adsorption/retention processes
I recommend to publish this paper in the present version.
Thanks for your comments

Reviewer 3 Report
References should be of the same type.
It is impossible and inappropriate for one article:
Countries like Argentina, China, India, Kenya, Mexico, 40 South Africa, Tanzania and Uganda (De La Cruz, Castillo, Arteaga, Cervantes, & Pinelo, 2013; 41 Valenzuela, Ramírez, Sol, & Reyes, 2011), suffer from fluorosis because of water consumption from 42 wells. Due to the high concentrations of fluoride ions reported in water, it has been necessary to 43 develop technologies that help eliminate fluoride ions from groundwater (De La Cruz et al., 2013).
At present, different methods have been used to improve the chemical and physical properties of 60 chitosan, mainly increasing its pore size, chemical, mechanical, and thermal resistance [10]. One of 61 the polymers used to improve these properties is polyvinyl alcohol (PVA), in conjunction with 62 chemical cross-linking or cross-linking to improve its resistance in acidic and basic solutions [9] - [12]. 63 The cross-linking of chitosan and PVA is a reaction with an agent that leads to the preservation of the 64 biopolymer through the formation of bonds between the chitosan and PVA chains. This process can 65 be carried out through covalent or ionic bonds. Covalent cross-linking is an irreversible chemical 66 reaction in which stable bonds are formed between the cross-linking agent and the polysaccharide 67 chains, allowing the chitosan to remain stable even in a very acidic medium (pH 3) [13].
Author Response
Reviewer 3
Dear reviewer, in attached file we send the correccones and suggested adjustments.
References should be of the same type.
It is impossible and inappropriate for one article:
Countries like Argentina, China, India, Kenya, Mexico, 40 South Africa, Tanzania and Uganda (De La Cruz, Castillo, Arteaga, Cervantes, & Pinelo, 2013; 41 Valenzuela, Ramírez, Sol, & Reyes, 2011), suffer from fluorosis because of water consumption from 42 wells. Due to the high concentrations of fluoride ions reported in water, it has been necessary to 43 develop technologies that help eliminate fluoride ions from groundwater (De La Cruz et al., 2013).
At present, different methods have been used to improve the chemical and physical properties of 60 chitosan, mainly increasing its pore size, chemical, mechanical, and thermal resistance [10]. One of 61 the polymers used to improve these properties is polyvinyl alcohol (PVA), in conjunction with 62 chemical cross-linking or cross-linking to improve its resistance in acidic and basic solutions [9] - [12]. 63 The cross-linking of chitosan and PVA is a reaction with an agent that leads to the preservation of the 64 biopolymer through the formation of bonds between the chitosan and PVA chains. This process can 65 be carried out through covalent or ionic bonds. Covalent cross-linking is an irreversible chemical 66 reaction in which stable bonds are formed between the cross-linking agent and the polysaccharide 67 chains, allowing the chitosan to remain stable even in a very acidic medium (pH 3) [13].
The references were corrected and the entire document was reviewed. The introduction was adjusted and more current references were attached. In the results and discussion section, references that support the results were also attached.

Round 2
Reviewer 1 Report
The manuscript has been improved but still needs minor revision.
In line 38 the change has not been made and cryolite is still not mentioned.
The order of citatations is inadequate. In line 37 there are references 1 and 2, followed by reference 7 line 38. It should be 3 instead of 7
References 3, 4, 5, 7, 10, 14, 15, 16, 19, 20: the name of the journaL is written incorrectly. It should be "Fluoride", instead of "Fluoride Research Report". This name does not exist. The name of the journal is correctly written in reference 6
In reference 64 "revista materia" should be replaced by "Revista Materia"
References 55 and 65: there are no names of the journals. In reference 65 "Adsortion" should be replaced by "Adsorption"
Figure 2. I have empty spaces instead of figure 2 on page 8.
In reference 43 the number of ISSN is not needed. It is also followed by inadequate format of citation
Author Response
Dear Reviewer
Thank you, all your comments have been made
In line 38 the change has not been made and cryolite is still not mentioned.
Fluorite, apatite, cryolite and micas are the most important fluorine-containing minerals
The order of citatations is inadequate. In line 37 there are references 1 and 2, followed by reference 7 line 38. It should be 3 instead of 7.
The change was made
References 3, 4, 5, 7, 10, 14, 15, 16, 19, 20: the name of the journaL is written incorrectly. It should be "Fluoride", instead of "Fluoride Research Report". This name does not exist. The name of the journal is correctly written in reference 6
The change was made
In reference 64 "revista materia" should be replaced by "Revista Materia"
The change was made
References 55 and 65: there are no names of the journals. In reference 65 "Adsortion" should be replaced by "Adsorption"
The change was made
Figure 2. I have empty spaces instead of figure 2 on page 8.
The change was made
In reference 43 the number of ISSN is not needed. It is also followed by inadequate format of citation
The change was made
